# Dissolution of Portlandite in Pure Water: Part 2 Atomistic Kinetic Monte Carlo (KMC) Approach

**DOI:** 10.3390/ma15041442

**Published:** 2022-02-15

**Authors:** Mohammadreza Izadifar, Neven Ukrainczyk, Khondakar Mohammad Salah Uddin, Bernhard Middendorf, Eduardus Koenders

**Affiliations:** 1Institute of Construction and Building Materials, Technical University of Darmstadt, Franziska-Braun-Str. 3, 64287 Darmstadt, Germany; koenders@wib.tu-darmstadt.de; 2Department of Structural Materials and Construction Chemistry, University of Kassel, Mönchebergstraße 7, 34125 Kassel, Germany; salahuddin@uni-kassel.de (K.M.S.U.); middendorf@uni-kassel.de (B.M.)

**Keywords:** portlandite, calcium hydroxide, atomistic kinetic Monte Carlo, upscaling approach, dissolution rate

## Abstract

Portlandite, as a most soluble cement hydration reaction product, affects mechanical and durability properties of cementitious materials. In the present work, an atomistic kinetic Monte Carlo (KMC) upscaling approach is implemented in MATLAB code in order to investigate the dissolution time and morphology changes of a hexagonal platelet portlandite crystal. First, the atomistic rate constants of individual Ca dissolution events are computed by a transition state theory equation based on inputs of the computed activation energies (Δ*G**) obtained through the metadynamics computational method (Part 1 of paper). Four different facets (100 or $\bar{1}00$, 010 or 0$\bar{1}$0, $\bar{1}10$ or $1\bar{1}0$, and 001 or 00$\bar{1}$) are considered, resulting in a total of 16 different atomistic event scenarios. Results of the upscaled KMC simulations demonstrate that dissolution process initially takes place from edges, sides, and facets of 010 or 0$\bar{1}$0 of the crystal morphology. The steady-state dissolution rate for the most reactive facets (010 or 0$\bar{1}0$) was computed to be 1.0443 mol/(s cm^2^); however, 0.0032 mol/(s cm^2^) for $\bar{1}10$ or $1\bar{1}0$, 2.672 × 10^−7^ mol/(s cm^2^) for 001 or 00$\bar{1}$, and 0.31 × 10^−16^ mol/(s cm^2^) for 100 or $\bar{1}00$ were represented in a decreasing order for less reactive facets. Obtained upscaled dissolution rates between each facet resulted in a huge (16 orders of magnitude) difference, reflecting the importance of crystallographic orientation of the exposed facets.

## 1. Introduction

To upscale atomistic simulations of mineral phases dissolution/precipitation towards much larger timescales and microscopic crystal sizes, kinetic Monte Carlo (KMC) simulations are applied using Molecular dynamics (MD) results from the Part 1 portlandite case study [1]. The principal idea of a KMC simulation is in bridging the timescale gap by coarse-graining the time evolution and focusing on discrete rare events using Markovian state-to-state dynamics [2]. With the application of such a promising upscaling tool, system dynamics at long timescales can be predicted by performing only short MD simulations [3]. At the atomistic scale, the interfacial properties are reflecting the chemical composition, type of bonds, crystallographic orientation of the exposed facets, impurities incorporated in the crystal, and lattice defects. At the mesoscopic level, the (pore) solution properties involve the nature of the solvent and its composition (i.e., saturation level), ionic strength, temperature, hydrodynamic conditions and other parameters. Piana et al. [4] carried out a 3D microscopic KMC simulation of a growing urea crystal in which the rate constants for corners and edge crystal sites were approximated by data from islands/steps on facets. Chen et al. [5] demonstrated the approach on NaCl crystal, by calculating the dissolution rate constants from ab initio MD simulations. Only NaCl (100) facets with different site types (e.g., edges and corners) were sufficient to perform a KMC simulation for the whole crystal due to crystal symmetry, high aqueous solubility and the absence of intramolecular degrees of freedom within the lattice. For more complex compounds, a simplified approach to efficiently determine the dissolution rates on the basis of crystal structure is proposed by Elts et al. [3].

The dissolution of calcium-silicate minerals is a fundamental process in cement hydration science, controlled by the intricate interplay of atomic crystal dissolution mechanisms such as spontaneous removal of numerous crystal sites that are activated at different energies. Moreover, at the mesoscopic scale, it also may depend on the saturation level of a pore solution, as well as the formation of facet complexes. However, to reveal the underlying origin of facets only would require upscaling of the atomistic modeling approaches. Initially, this is to be achieved at far-from-equilibrium conditions, which have only recently begun to be explored. This will enable atomistic prediction of the dissolution rate as a function of individual (atomistic) activation energies, as it arises only from the mineral nanoscale topography, thus, to clearly separate it from mesoscale effects of, e.g., interplay with pore solution concentrations, i.e., the free energy driving force (Δ*G*) between mineral and solution. Therefore, the mechanism of ionic (calcium silicate) minerals dissolution deserves more in-depth research, as it is the critical missing link that would unlock a fundamental understanding of cement hydration.

Recently, the atomistic kinetic Monte Carlo (KMC) model was proposed by Martin et al. [6] for the dissolution of a simple Kossel crystal, which successfully described the experimentally observed sigmoid dependence of the dissolution rate versus the free energy driving force (Δ*G*). This major achievement was made possible by invoking the reversibility of the chemical reactions at the microscopic level. The new KMC model confirmed that the onset of the rapid dissolution rate originates from the opening of pits, which constantly supply terraces for step retreat. However, the two main parameters of the simplified atomistic model, namely the dissolution and precipitation energies, have been calibrated to correctly fit the dissolution rates of several representative aluminosilicate minerals, as well as alite. Thus, the next step is to downscale the recent KMC approach from Martin et al. [6], by going beyond the most recent state-of-the-art implementation based on over-simplified Kossel crystals and considering more realistic specifics of the calcium(-silicate) based crystal structures. For this, a direct link between the atomistic simulations and KMC should be established. Recently, Martin et al. [7] successfully described the dissolution of a quartz crystal by the atomistic kinetic Monte Carlo model, confirming measurement-observed dependence of the dissolution rate as a function of free energy driving force, which has a transition state theory behavior in contrast to an empirical sigmoidal dependence observed for aluminosilicate minerals.

Mixing cement with water results in cement dissolution and subsequent precipitation of portlandite, calcium silicate, and calcium aluminate hydrates which bind sand and gravel aggregates in concrete. Portlandite or calcium hydroxide (Ca(OH)_2_) is a significant mineral phase precipitated as hexagonal crystals of various sizes filling the pore space within the bulk cement paste and more porous interfaces around inert aggregates. Portlandite plays a substantial role in the mechanical and durability properties of cement paste [8] as well as corrosion protection [9]. Other widely exerted aspects include dewatering sludge [10,11], improving the mechanical properties of fly ash cement [12,13], delaying steel corrosion [14], resistance to leaching and degradation in water [8,15,16] and acids [17,18]. The Ca(OH)_2_ is also very important in other areas of chemical industry, so the knowledge generated here is of general importance.

In Portland cement paste, portlandite represents the second most abundant hydration phase, with a fractions of about 15–25%, but presents the most soluble hydration phase [8]. Thus it is the phase that dissolves first and creates porosity in case of leaching in pure water, or exhibits an intense reactivity to the CO_2_ results in the entire re-crystallization of portlandite into the calcite [19,20]. The high initial pH of the Portland cement paste (~14) is typically higher than for saturated portlandite solution (12.5) due to presence of soluble alkalis (common ion effect). This pH is significantly reduced with carbonation or other acid attack reactions due to the dissolution of first portlandite and then other less soluble hydrates, resulting in increased porosity and degradation of concrete and imposing mechanical and durability issues for reinforced concrete structures [21].

Hence, the main objective of this study is to develop an elementary physical/chemical bridging model for the initial dissolution of portlandite hexagonal crystals. Portlandite is proposed here, as the simplest benchmark representative of cementitious mineral phases, for the long-sought goal of connecting the nanoscale to the upscaled microscale level. To understand the effects of equilibrium crystal morphology of portlandite [22] during the dissolution process, different facets of 100 or $\bar{1}00$, 001 or 00$\bar{1}$, 010 or 0$\bar{1}0$, and $\bar{1}10$ or $1\bar{1}0$ were considered according to the Wulff construction reported by Chen et al. [5]. Then, the far-from-equilibrium kinetic Monte Carlo (KMC) approach was employed to investigate the upscaling atomistic dissolution rates of portlandite, representing the forward reaction rate. Moreover, to implement the KMC approach in a proper and accurate way, Salah Uddin et al. in parallel study (Part 1) [1] provided input information about the reaction activation energy (Δ*G**) of the dissolution of calcium atoms (Ca) for different neighbor scenarios and for different facets (crystal planes or surface orientations) at room temperature by a molecular dynamic (MD) [23,24] computational method using ReaxFF coupled with a metadynamics approach as input values to compute the dissolution rates (r_D_) according to transition state theory (Equation (1)). In fact, in the KMC upscaling approach, the dissolution and precipitation activation energies for a given site are sometimes written as the sum of the contribution of the n bonded neighbors [2]. Then, by application of a MATLAB code for a KMC upscaling approach, the mesoscopic (total) dissolution rate is computed based on time coarse-graining and probabilistic evolution of dissolution rates of individual atomistic events already computed for different scenarios by the Arrhenius-like equation of the TST.
(1)$$k_{D}=\frac{k_{B}T}{h}\;exp^{-\frac{\Delta G*}{RT}}$$
where $k_{B}$ is the Boltzmann constant, h Planck’s constant, Δ*G** is the free energy of activation calculated from meta D simulations (inputs from Part 1 paper) [1], R is the gas constant and T is the temperature.

## 2. Methods and Computational Models

### 2.1. Quantum Chemistry Computations

Our intention in this section is to investigate the enthalpy (Δ*Ha*) at 0 K through the nudged elastic band (NEB) by DFT computational approach in order to decipher the relative difference in behavior with the computation of total activation energy (Δ*G**) obtained from metadynamics computational approach at 298 K for dissolution of Ca at the transition state. Therefore, density functional theory (DFT) [25,26,27,28,29,30] computational approach was employed as defined in the Vienna ab initio simulation package (VASP) [31,32,33] for the computation of the electronic structure. The Perdew–Burke–Ernzerhof (PBE) functional was exerted to explain the electron exchange and correlation energy within the generalized gradient approximation [34]. A well-converged plane-wave cutoff energy of 400 eV was employed. The projector-augmented wave method and pseudopotential were used to describe the electron-ion interaction [35]. A force tolerance of 10^−2^ eV Å^−1^ was applied for structural optimizations. The break condition of 10^−4^ eV was considered for the convergence of electronic self-consistent loop. The Brillouin zone was sampled using a well-converged k-sampling equivalent given by 1 × 1 × 1 Monkhorst-Pack k-points for the total system [36]. The initial lattice parameters of portlandite are as follows: a = 3.585 Å, b = 3.585 Å, c = 4.871 Å, α = 90°, β = 90°, and γ = 120°. A three-dimensional visualization software for electronic and structural analysis (VESTA) was also used to display the crystalline structure of our modeling [37].

The nudged elastic band (NEB) method [38,39] is a chain-states method at atomic-scale for calculating minimum energy pathway and finding transition state between a reactant and a product sates. The minimum energy pathway represents how the atoms would evolve (between given the initial and the final states), and where a maximum in the potential energy along that path represents the activation energy of the studied process. Initially, the method is implemented, as the geometries of the initial and the final systems are optimized to minimize their energy. Then a rough approximation of the reaction pathway is built, a set of images are created by performing a linear interpolation between the initial and final systems. An intermediate system can be provided, in which case the interpolation is performed between the initial and intermediate systems, and then between the intermediate and final systems. Finally, a reaction path is found by performing within a simultaneous optimization of all images. Figure 1 illustrates the minima energy procedure of Ca dissolution on the 001 or 00$\bar{1}$ facets for the first scenario in the presence of all neighbors through NEB by the DFT computational method. As can be observed, the energy of the portlandite system at the transition state (c) for Ca dissolution increased almost 2.51 eV, which equals to 242.18 (kJ/mol). The enthalpy (Δ*Ha*) for activation energy obtained by DFT computational method corresponds to T = 0 K, which may demonstrate the importance of the entropy (almost 110 kJ/mol) contribution to the total activation energy, obtained by MD computation (Table 1). In other words, contribution of entropy (Δ*S**) could be (roughly) estimated by the difference between the total activation energy (Δ*G**) at 298 K, and the enthalpy (Δ*Ha*) at 0 K computed through metadynamics and DFT computational approaches, respectively.

### 2.2. Atomistic Kinetic Monte Carlo

In order to implement the Atomistic Kinetic Monte Carlo (KMC) upscaling approach for dissolution of portlandite in the aqueous ambient atmosphere, a MATLAB code was developed to compute the time of dissolution of portlandite for a supercell consisting of 83,629 atoms and 17,461 sites. To execute the MATLAB code, it was initially needed to compute the dissolution rate constant of Ca for seven various scenarios depending on the existing neighbors for the facets of 001 or 00$\bar{1}$; moreover, three different scenarios for the facets of 100 or $\bar{1}00$, 001 or 00$\bar{1}$, 010 or 0$\bar{1}0$, and $\bar{1}10$ or $1\bar{1}0$ as shown in Figure 2 and Figure 3, respectively. The reason for choosing different scenarios (also called atomistic events) for the computation of each atomistic Ca dissolution rate is due to the effects of the neighbors on the computation of activation energy during the dissolution of each particular Ca. In this way, activation energy of Ca for each scenario has been calculated in order to compute the dissolution rate constant (Equation (1)). It should be noted that in Figure 3, there are 3 neighbors Ca, left, right (shown with green and red colors), and behind (not shown in the Figure 3).

The following process briefly explains how to implement the KMC [40] algorithm for each dissolution site selection and dissolution time advancement through developed MATLAB code. Initially, 16 different possible events regarding four different facets must be tabulated with their dissolution rates. Then, it is needed to compute the total rate constant (*k_tot_*) of crystal morphology for all the exposed sites to the environment matching with their possible events. For each time iteration, it is needed to update the book-keeping of all facet sites (which are not in bulk) of crystal after dissolution of each site to identify the newly exposed facet sites. To compute the total rate constant (*k_tot_*) all facet sites are added/summed up (Equation (2)).
(2)$$k_{tot}=\overset{N_{p}}{\underset{p=1}{\sum }}k_{p}$$

The probability of each event between 0 to 1 (16 possible events for all different scenarios) is computed by normalizing the rate of each event, which is multiplied by the number of sites and then divided by the *k_tot_* (Equation (3)).
(3)$$K_{i,p}=\frac{k_{i,p}}{\sum_{p=1}^{N_{p}}\sum_{i=1}^{N_{L}}k_{i,p}}=\frac{k_{i,p}}{\sum_{i=1}^{N_{L}}k_{i,tot}}=\frac{k_{i,p}}{k_{tot}}$$

Then, a random number of $\zeta_{1}$ between 0 to 1 is then generated to select the probability of occurring event (Equation (4)) and, consequently, random selection of the site by an integer number from that event. Finally, the time of selected site for dissolution is then computed by division of second random number of $\zeta_{2}$ between 0 to 1 with *k_tot_* (Equation (5)).
(4)$$\;\;\;\;\;\overset{I}{\underset{i=1}{\sum }}\frac{k_{i,p}}{k_{tot}}\geq \zeta_{1}\geq \overset{I}{\underset{i=1}{\sum }}\frac{k_{i,p}}{k_{tot}}$$
(5)$$\Delta t=-ln\left(\zeta_{2}\right)/k_{tot}$$

### 2.3. The MATLAB Code Implementation by Employing the KMC Upscaling Approach

The coding implementation can be split in four major sections: (1) pre-processor, (2) event processor, (3) solver and (4) post-processor. Pre-processor prepares and reads the input (crystal) structure, i.e., creates a list of positions for all particles. A supercell crystal morphology of portlandite, which has been prepared beforehand must be imported into the MATLAB code. Firstly, in order to enable easy tracking of the atoms during the dissolution process, the position of each site was indexed. Atom positions are defined by parameters in cartesian coordinate system for three different horizontal axes of 60°, 90°, 120°, and a vertical axis of Z. Secondly, it is needed to separate the sites into two groups: (1) inner sites (blocked) and (2) outer sites (side/surface, available for dissolution). The reason for the separation of sites is due to the dissolution process, which is only carried out on the outer sites (side/surface) being exposed to the sounding (pure water) solution environment. Outer sites can be separated into three different sides such as 60°, 90°, 120°, and a surface, based on the four different facets for crystal morphology of portlandite. Event processor executes individual events, keeps track of them at the system level, and performs updates after changes in the system. Here, it is needed to update the inner sites to become a part of side/surface, e.g., after the process of dissolution of neighbors from the outer sites (side/surface). It is also worth mentioning that the rate of dissolution for each site depends on the number of (missing) neighbors, which must be defined for all three different sides and the surface. Therefore, according to the crystal morphology of portlandite, seven different events for the surface and three different events for each side can be considered, i.e., in total sixteen different events for the dissolution process. Thirdly, the solver implements the actual rejection-free KMC algorithm for probabilistic-based selection of an event to execute. Based on the (16) dissolution rate constants (k) for all sides and surface as shown in Table 1, Table 2, Table 3 and Table 4; and moreover, according to the theory of KMC upscaling approach as described in Section 2.2, the search for finding the most probable outer site to be dissolved is performed. Fourthly, after the dissolution of one site, it is essential to inform the neighbors about the missing of it. This will automatically change the blocked sites into an outer (unblocked) site, which can thus contribute to the list of possible processes to be selected for dissolution (in the next time step). Finally, the third and fourth processes must be continued consecutively until reaching a user defined number of (time) steps, or else all sites have been dissolved. In post-processor, the data are prepared for exporting and plotting purposes.

## 3. Results and Discussions

According to the activation energy (Δ*G**) obtained through a metaD computational method by Salah Uddin et al. (Part 1 of companion paper) regarding Ca dissolution for all different scenarios, the dissolution rate constants were initially computed for all possible scenarios using Equation (1). In total, 16 different Ca atom dissolution events (MD simulations) were used, including the facets of 001 or 00$\bar{1}$, 100 or $\bar{1}00$, 010 or 0$\bar{1}0$, and $\bar{1}10$ or $1\bar{1}0$ as shown in Table 1, Table 2, Table 3 and Table 4, respectively. According to the DFT computational method, obtained activation enthalpy (Δ*Ha* in Table 1) of all seven different scenarios on the 001 or 00$\bar{1}$ facets showed a decreasing trend as the number of neighbors decreased. Then, the difference between the activation energy (Δ*G**) computed through metaD and enthalpy (Δ*Ha*) through DFT methods demonstrates the contribution of entropy for total activation energy calculation, which might be significant and should be considered for atomistic rate constant computations. Here it should be noted that the DFT calculations represent at 0 K temperature, while the MD simulations are run at room temperature (298 K). The absolute contribution of entropy was difficult to quantify, as it is challenging to be obtained for individual atomistic events. Unlike typically employed approach for equilibrium reaction thermodynamics, where the change of enthalpy with temperature could be calculated (by knowing the heat capacity dependency with temperature) according to the Arrhenius-like equation of the TST, this methodology is formidable for individual atomistic events. However, the comparison between Δ*Ha* (DFT) and Δ*G** (metaD) results shows very good qualitative agreement, capturing the same trends as a function of neighbor’s configurations. The change in relative difference, calculated as rel_error = (Δ*Ha* − Δ*G**)/Δ*G**, was quite consistent, with a maximum and minimum rel_error of −35.5% and −46%, a mean value of −45%, and a standard deviation of this mean being only 4% (absolute error value). This indicates that a consistent increase (i.e., shift) of all ΔHa values for a mean relative difference (+45%) would result in a 4% error. This shift is due to a combined effect of (a) entropy and (b) enthalpy increase due to temperature increase (from 0 to 298 K). More research in this direction is needed in order to obtain (Δ*G**) with contribution of entropy directly from DFT computational approach. For this, the use of vibration frequencies to separate entropy contributions in TST equations seems to be an elegant approximation [7].

Next, it was attempted to test a correlation between the seven scenarios of Figure 2 and the resulting dissolution rate constants (Table 1). According to Elts et al. [3], the logarithmic values of the rate constants *y* = ln(*k*_i_) can be fitted as a function of the number of neighbors (*x_i_*) with an Allometric (basic Origin) function *y_i_* = *a* + *bx_i_^c^*, resulting in *a* = 10.9, *b* = −0.581 and *c* = 2.98 as calibrated parameters. The correlation resulted in a quite convincing adjusted R-square coefficient of 0.932, indicating that the selected model follows the trend of the data satisfactorily. Rewriting this model in exponential form and comparing it to the Arrhenius equation, one obtains the direct dependence Δ*G**(*x*) = −*bRTx^c^*.

After computation of the rate constants for all 16 different scenarios applied on all facets, a KMC MATLAB code was developed to compute the time of site dissolution for the whole crystal morphology of portlandite consisting of 17,461 sites as shown in Figure 4.

Figure 5 shows site-by-site dissolution model of portlandite after 5250 (a,d), 10,500 (b,e), and 15,750 (c,f) steps. Each step is representative of one site dissolution. The dissolution process for the crystal morphology of portlandite is basically performed from 010 or 0$\bar{1}0$ (facet 60°), and $\bar{1}10$ or 1$\bar{1}0$ (facet 90°) facets. The results show that the dissolution process of the crystal is mostly happening for the scenarios of 010 or 0$\bar{1}$0 facets, the common sides with facets of $\bar{1}10$ or 1$\bar{1}0$ (medium red color as shown in Figure 4), and the common edges with 001 or 00$\bar{1}$ (dark red color as shown in Figure 4). This is due to the greater value for the event probability according to the greater computed rate constants. On the other hand, as the process of sites dissolution progress from 010 or 0$\bar{1}$0 facets, a small contribution of $\bar{1}10$ or $1\bar{1}0$ facets account for less than one to four percent regarding the first 5250 sites dissolution to the dissolution of the whole crystal as shown in Figure 6, respectively. This is due to the formation of sites with missing left and right neighbors for $\bar{1}10$ or $1\bar{1}0$ facets; and consequently, the higher chance of event probability being selected because of its large rate constant (4.643 × 10^12^). It must be highlighted that after several numerical realizations (at least 10 times) of KMC MATLAB code for sites dissolution of portlandite crystal, contributions of 100 or $\bar{1}00$, and 001 or 00$\bar{1}$ facets have not been observed at all. In other words, not one site dissolved in the simulated time frame. Slow reacting facets were simulated separately, to deal with the huge time differences to dissolve similar number of atoms (results show later). The only variability in the numerical realizations was observed for the contribution of $\bar{1}10$ or 1$\bar{1}0$ (facet 90°) facets, resulting in 2% (8 numerical realizations, Figure 6c) and 3% (2 numerical realizations), and 4% (8 numerical realizations, Figure 6d) and 5% (2 numerical realizations) after 15,750, and 17,461 steps, respectively.

The dissolution time of each individual site along 5250 (a), 10,500 (b), 15,750 (c), and 17,461 (d) steps were shown in Figure 7. It is clear that the majority of sites from both facets of 010 or 0$\bar{1}0$ (green point), and $\bar{1}10$ or 1$\bar{1}0$ (purple cross) have been dissolved between 10^−11^ to 10^−14^ seconds as shown in (a–c). Those sites which have been dissolved for the time less than 10^−14^ second are concerning to the larger second random number selection which is close to 1 for dissolution time computation. However, the trend of random number selection is uniform, and it is impossible to avoid those larger random number selections close to 1.

On the one hand, Figure 8 has been plotted to illustrate the time evolution of the total number of dissolved sites after 5250 (a), 10,500 (b), 15,750 (c), and 17,461 (d) steps. The slope of the line stays monotonic from almost 2000 to 15,750 sites dissolution (a–c). This is due to the almost identical total rate constant (*k*_tot_) for each step computation, resulting in little change of the statistical average of a dissolution kinetics (inverse of slope in Figure 8). In contrast, the slope of the total number of dissolved sites for the first 2000 sites behaved differently, i.e., exhibiting a dynamical increase in the averaged dissolution rate within the region highlighted with the green ellipse in Figure 8a. At the beginning of the dissolution process, the event probability to be selected for site dissolution on the edges, sides, and facets of 010 or 0$\bar{1}$0 (facet 60°) due to the Ca atoms having more neighbors (e.g., Ca with one or two neighbors, Table 3 scenarios a and b). As time goes on with more site dissolution, the formation and then contribution of scenario c with no neighbors on the facets of $\bar{1}10$ or 1$\bar{1}0$ (facet 90°) gradually appears, which has a higher probability to be dissolved due to the greater rate constant. It is also worth mentioning that for the remaining 1711 sites, the dissolution time for 010 or 0$\bar{1}0$ (facet 60°) facets increased (between 10^−11^ to 10^−9^) in comparison to $\bar{1}10$ or 1$\bar{1}0$ (facet 90°) facets (between 10^−12^ to 10^−15^) as shown with the green and purple ellipses in Figure 7d, respectively. In other words, the contribution of scenario c from facets of $\bar{1}10$ or 1$\bar{1}0$ can increase the average dissolution rate (i.e., decrease the time of dissolution per step) as shown with the purple ellipse in Figure 7d because of greater rate constant in comparison to the scenarios a and b of 010 or 0$\bar{1}0$ facets. Therefore, the slope of the line changes as shown with the green ellipse in Figure 8d.

Figure 9 shows the time evolution of sites dissolution of portlandite for each facet independent from contributions of other facets. From the slope of the curve in Figure 9b, the steady-state dissolution rate for the most reactive facets (010 or 0$\bar{1}0$) was obtained to be 67.2 × 10^10^ s^−1^ (i.e., the number of Ca atoms per second, for the specific case simulation). In more common mole units and normalized to the facet area, this can be recalculated to be 1.0443 mol/(s cm^2^) by considering the Avogadro’s constant and that the initial facet area of the most reactive (010 and 0$\bar{1}0$) facets is in total 106.86 nm^2^ (=6.094 nm × 8.768 nm × 2 facets). By considering the full crystal with two basal facets and all six lateral (rectangular) facets, the rate would be an order of magnitude lower (normalized to larger facet area).

Evolution of dissolution rates (i.e., slopes) for the rest of facets is shown in Figure 9. Obtained steady-state dissolution rates are in decreasing order is as follows: 0.0032 mol/(s cm^2^) for $\bar{1}10$ or $1\bar{1}0$ (c), 2.672 × 10^−7^ mol/(s cm^2^) for 001 or 00$\bar{1}$ (a), and 0.31 × 10^−16^ mol/(s cm^2^) for 100 or $\bar{1}00$ (d), facets.

Such huge differences in dissolution rates for the different facets were already discussed in Part 1 of paper [1]. At the atomistic scale, the interfacial properties are reflecting the chemical composition, type of bonds, and crystallographic orientation of the exposed facets, impurities incorporated in the crystal, and lattice defects. However, at the mesoscopic level, the solution properties involve the nature of the solvent and its composition (saturation level), ionic strength, temperature, hydrodynamic conditions and other parameters.

Finally, it is interesting to attempt an (ambitious) comparison with available macroscopic measurements from literature [41], which reported a value of 5.40 × 10^−8^ mol/(s cm^2^). It should be noted that to be able to make such a comparison a huge gap needs to be further upscaled in order to reach the macroscopic predictions. In experimental measurements the dissolution of portlandite was found to be controlled by the ion diffusion process through stagnate hydrodynamic interface layer. Therefore, the comparison of the KMC upscaling approach with experimental data is still very limited and requires updates on both experimental (downscaling) and modeling (upscaling) sides. On the modeling side, the atomistic KMC should incorporate the effect of solution concentration on atomistic processes and be coupled with continuum-based models to capture the concentration gradients at the interface. The solution in a hydrodynamic boundary layer is expected to have a much higher (Ca^2+^ and OH^−^) ion concentrations, than as simulated by our fully diluted (forward reaction rate) case. Realistic experimental may result in some precipitations (backward reaction) on the measured portlandite facet that would significantly reduce the dissolution rate [7]. Moreover, experimental measurement was undertaken on pressed portlandite powder tablets polished with 1 um spray; thus, the facet roughness and porosity, being far from perfect, further signify diffusion mechanisms. Moreover, the information of the crystal morphology and facets distribution available at the disk facet was not reported in Wang et al. [41]. Certainly, the real fractional distribution of each facet being exposed to the solution interface is another (statistical) parameter that would enable direct prediction of the overall rate, and thus compare it with macroscopic measurements.

## 4. Conclusions

The comparison between Δ*Ha* (DFT) and Δ*G** (metaD) results showed very good qualitative agreement, capturing same trends as a function of neighbor configurations. The Δ*Ha* trend had a consistent shift for a mean relative difference of −45% +/−4%, due to the combined effect of entropy and temperature (0 vs. 289 K).

For KMC, 16 different atomistic scenarios for Ca dissolution were considered depending on the existing neighbors for facets of 001 or 00$\bar{1}$, as well as three different facets of 100 or $\bar{1}00$, 001 or 00$\bar{1}$, 010 or 0$\bar{1}0$, and $\bar{1}10$ or $1\bar{1}0$. This allowed the computation of KMC for upscaling of the atomistic rate constants of the different scenarios into mesoscale rate and enabled the visualization of the evolution of crystal morphologies during the dissolution process (Figure 3 and Figure 4).

The results showed that the facets of 001 or 00$\bar{1}$, and 100 or $\bar{1}00$ represented a very small dissolution rate constant, which allowed two scenarios from 010 or 0$\bar{1}$0 and one scenario from $\bar{1}10$ or 1$\bar{1}0$ facets to contribute to the dissolution process. Moreover, the upscaled dissolution rate follows a linear trend from almost 2000 to 15,750 sites, and finally the upscaled rate of site dissolution for 010 or 0$\bar{1}0$ (facet 60°) facets decreased due to the reduction of the facet sites for the computation of the total rate constant.

The steady-state dissolution rate for the most reactive facets (010 or 0$\bar{1}0$) was reported to be 1.0443 mol/(s cm^2^), while 0.0032 mol/(s cm^2^) for $\bar{1}10$ or $1\bar{1}0$, 2.672 × 10^−7^ mol/(s cm^2^) for 001 or 00$\bar{1}$, and 0.31 × 10^−16^ mol/(s cm^2^) for 100 or $\bar{1}00$ were obtained in a decreasing order for less reactive facets. These results are important for a general understanding of the cement hydration and other chemical processes with portlandite.

## Figures and Tables

**Figure 1 materials-15-01442-f001:**
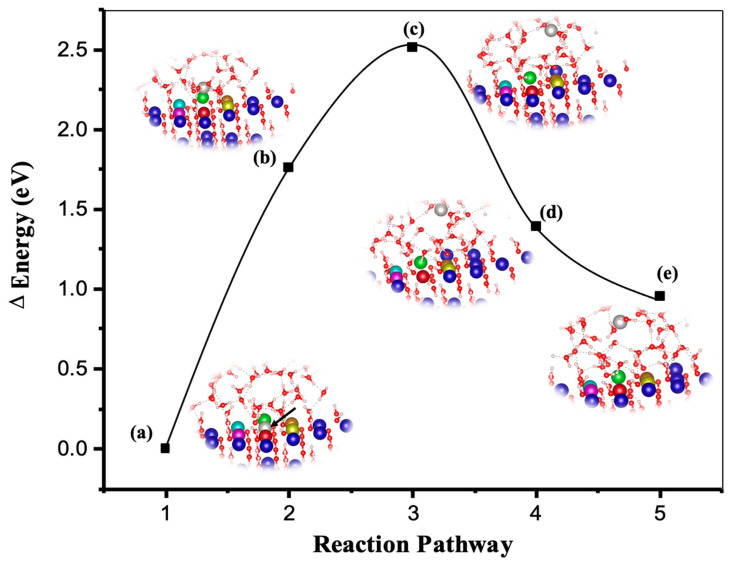
The DFT calculation of minimum energy pathway for Ca (light gray atom as shown with the arrow in the ground-state structure of (**a**)) dissolution on the 001 or 00$\bar{1}$ facets for the first scenario in the presence of all neighbors. (**a**,**e**) show the initial and final ground-state structures, (**b**–**d**) also illustrate the energy pathway of portlandite as the Ca dissolved.

**Figure 2 materials-15-01442-f002:**
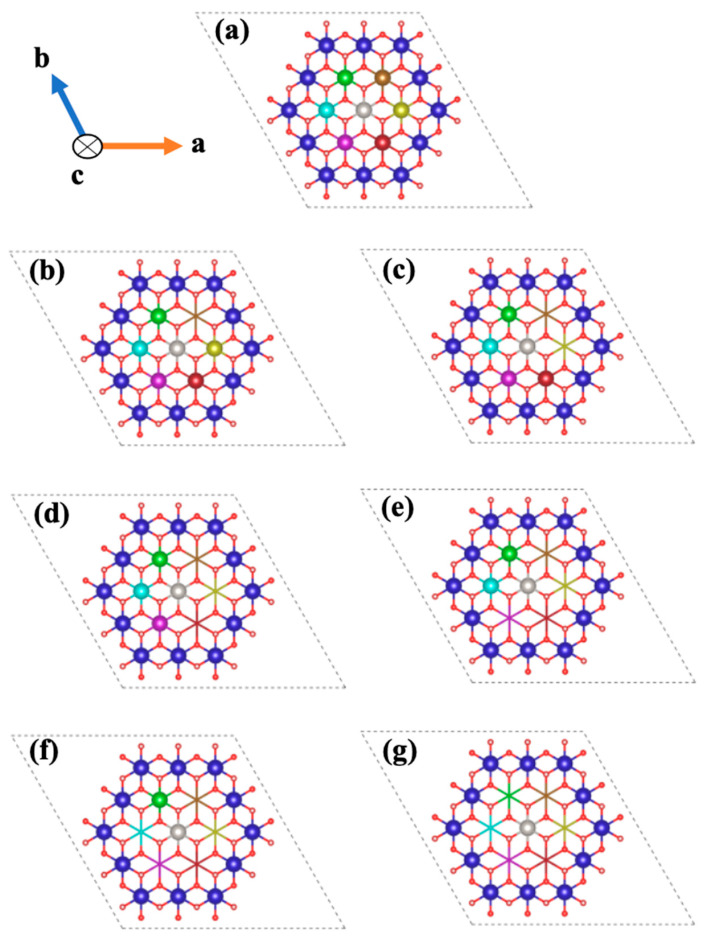
Illustration of seven different scenarios for the central (light gray) Ca atom dissolution depending on the possible significant configurations of existing nearest neighbors (in the presence of all 6 nearest neighbors (**a**), 5 neighbors (**b**), 4 neighbors (**c**), 3 neighbors (**d**), 2 neighbors (**e**), 1 neighbor (**f**), and in the absence of all nearest neighbors (**g**)) on the 001 or 00$\bar{1}$ facets of portlandite. Resulting scenarios are proposed to feed the KMC upscaling approach.

**Figure 3 materials-15-01442-f003:**
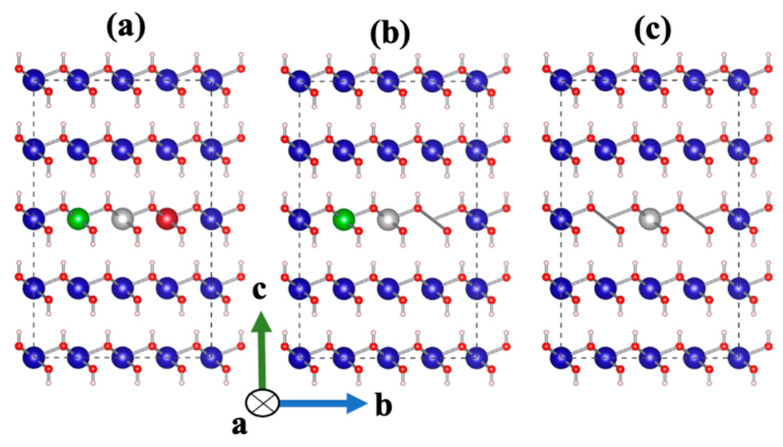
Illustration of three different scenarios for light gray Ca dissolution depending on the existing neighbors (in the presence of both left and right neighbors (**a**), in the presence of just left or right neighbor (**b**), in the absence of both left and right neighbors (**c**)) on the 100 or $\bar{1}00$, 010 or 0$\bar{1}0$, and $\bar{1}10$ or 1$\bar{1}0$ facets of portlandite employing KMC upscaling approach.

**Figure 4 materials-15-01442-f004:**
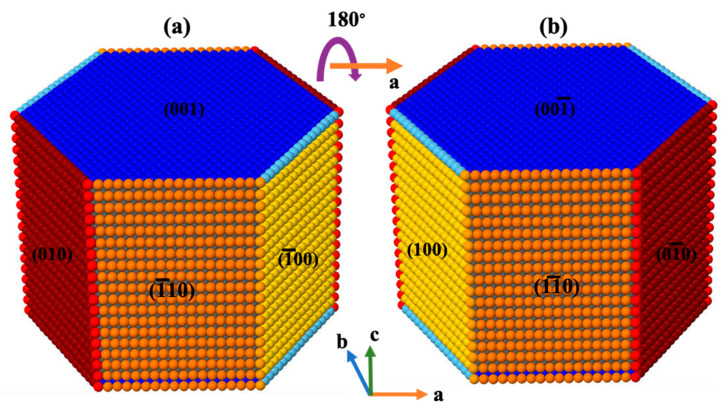
Snapshots of the crystal morphology of portlandite consisting of 17,461 sites from two various perspectives (dimensions of lateral rectangle are 6.094 nm × 8.768 nm). For better visualization of each facet, they were shown with different colors, while atoms with less neighbors have lighter color tone. Figure on the right side (**b**) represents a 180-degree rotation about (**a**) axis of the crystal on the left side.

**Figure 5 materials-15-01442-f005:**
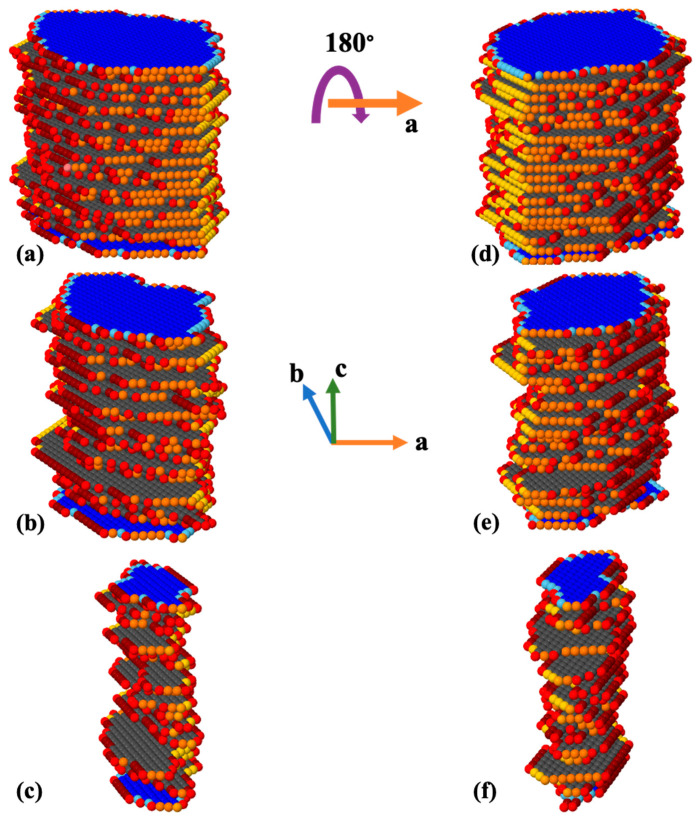
Evolution of calcium hydroxide crystal morphology during an atom-by-atom site dissolution KMC simulation. Initial crystal (Figure 4) consists of 17,461 sites, and after 5250 (**a**,**d**), 10,500 (**b**,**e**), and 15,750 (**c**,**f**) dissolution steps. Each step is representative of one site dissolution. Figures on the right side (**d**–**f**) represent a 180-degree rotation about (**a**) axis of the crystals on the left side (**a**–**c**).

**Figure 6 materials-15-01442-f006:**
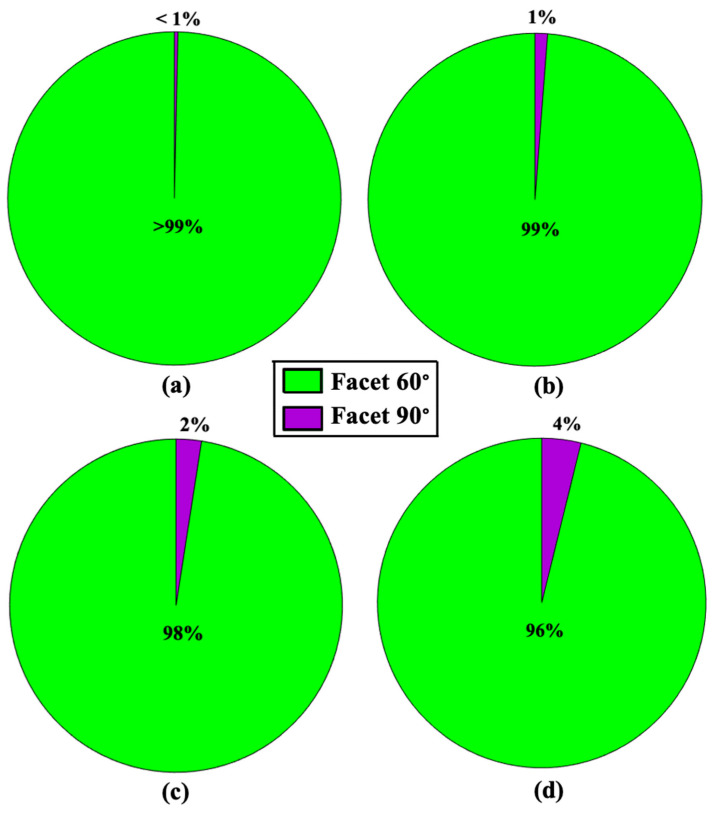
Contribution of each facet during sites dissolution of the whole crystal morphology of portlandite, after 5250 (**a**), 10,500 (**b**), 15,750 (**c**), and 17,461 (**d**) steps. Each step is representative of one site dissolution. Green and purple colors show the percentage of sites dissolution, which took place from 010 or 0$\bar{1}0$ (facet 60°), and $\bar{1}10$ or 1$\bar{1}0$ (facet 90°) facets, respectively.

**Figure 7 materials-15-01442-f007:**
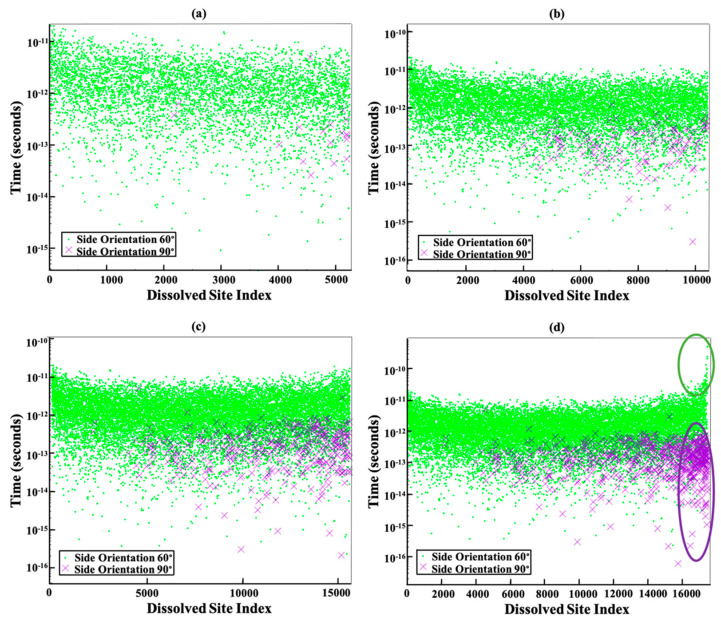
Each point or cross represents the dissolution time for each site along 5250 (**a**), 10,500 (**b**), 15,750 (**c**), and 17,461 (**d**) steps. Each step is representative of one site dissolution. Green point and purple cross show the time of each site dissolution, which take place from 010 or 0$\bar{1}0$ (facet 60°), and $\bar{1}10$ or 1$\bar{1}0$ (facet 90°) facets, respectively.

**Figure 8 materials-15-01442-f008:**
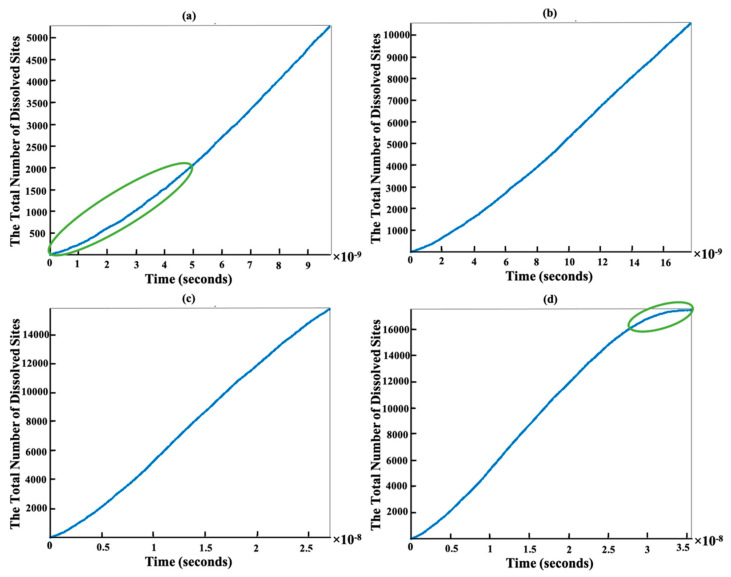
Time evolution of sites dissolution for the crystal morphology consisting of 17,461 sites during 5250 (**a**), 10,500 (**b**), 15,750 (**c**), and 17,461 (**d**) dissolution steps. Each step is representative of one site dissolution.

**Figure 9 materials-15-01442-f009:**
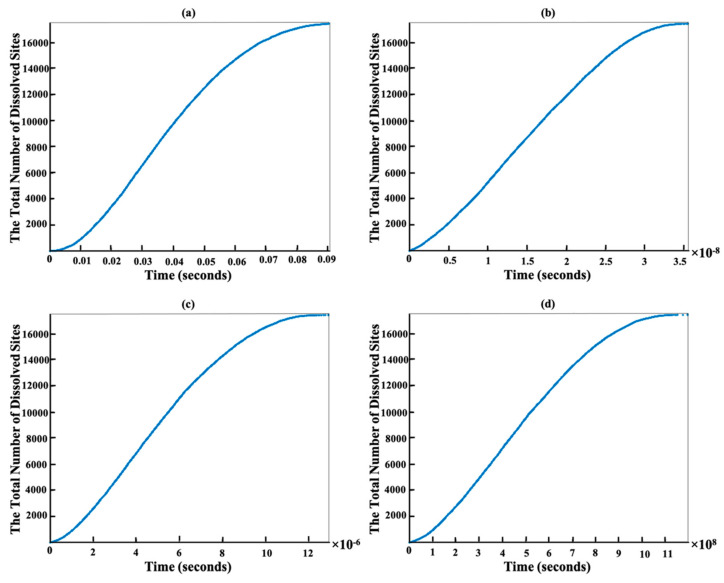
Time evolution of sites dissolution of the whole crystal morphology of portlandite for each facet independent from contribution of other facets. 001 or 00$\bar{1}$ (**a**), 010 or 0$\bar{1}0$ (**b**), $\bar{1}10$ or $1\bar{1}0$ (**c**), and 100 or $\bar{1}00$ (**d**).

**Table 1 materials-15-01442-t001:** Activation energy (Δ*G**) and enthalpy (Δ*Ha*) computations of light gray Ca dissolution on the facets of 001 or 00$\bar{1}$ for seven different scenarios at the room temperature (298 K) as shown in Figure 2 by molecular dynamics and density functional theory (DFT) simulation methods, respectively. Rate constants of light gray Ca dissolution computed according to the activation energies obtained through molecular dynamic simulation method at the room temperature.

Figure 2	(a)	(b)	(c)	(d)	(e)	(f)	(g)
Δ*G** (kJ/mol)	352.00	199.10	175.40	56.14	55.80	54.90	25.90
Δ*H_a_* (kJ/mol)	242.18	140.87	120.12	41.43	40.85	39.87	18.32
*k* (s^−1^)	1.243 × 10^−49^	7.849 × 10^−23^	1.119 × 10^−18^	0.897 × 10^3^	1.029 × 10^3^	1.479 × 10^3^	1.791 × 10^8^

**Table 2 materials-15-01442-t002:** Activation energy (Δ*G**) and dissolution rate constant (*k*) computations of light gray Ca dissolution from 100 or $\bar{1}$00 facets at the room temperature according to Figure 3.

Figure 3	(a)	(b)	(c)
Δ*G** (kJ/mol)	195.30	114.60	70.00
*k* (s^−1^)	3.638 × 10^−22^	5.081 × 10^−8^	3.337

**Table 3 materials-15-01442-t003:** Activation energy (Δ*G**) and dissolution rate (*k*) computations of light gray Ca dissolution from 010 or 0$\bar{1}0$ facets at the room temperature according to Figure 3.

Figure 3	(a)	(b)	(c)
Δ*G** (kJ/mol)	29.90	20.55	7.1
*k* (s^−1^)	3.565 × 10^7^	1.552 × 10^9^	3.536 × 10^11^

**Table 4 materials-15-01442-t004:** Activation energy (Δ*G**) and dissolution rate (*k*) computations of light gray Ca dissolution from $\bar{1}10$ or 1$\bar{1}0$ facets at the room temperature according to Figure 3.

Figure 3	(a)	(b)	(c)
Δ*G** (kJ/mol)	59.99	34.91	0.72
*k* (s^−1^)	0.189 × 10^3^	4.720 × 10^6^	4.643 × 10^12^

## Data Availability

Data is contained within the article.

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
