# Peer review of "Dissolution of Portlandite in Pure Water: Part 2 Atomistic Kinetic Monte Carlo (KMC) Approach"

_materials, 2022, doi:10.3390/ma15041442_

Round 1

Reviewer 1 Report

The study is conducted to explore an elementary physical/chemical bridging model for the initial dissolution of portlandite hexagonal crystals. The paper can be accepted after the minor modification.

The abstract is too long, which should be concentrated.

The convergence criterion of energy for each atom in the DFT simulation should be present.

The MATLAB code to implement the Atomistic Kinetic Monte Carlo method can be introduced in detail.

The following works about DFT and MD simulations can be cited if could: Adv. Funct. Mater. 2022, 2110846; Nanomaterials 2021, 11, 2236; Phys. Chem. Chem. Phys., 2021, 23, 24915.

Reviewer 2 Report

See atached pdf file.
